# Predictors and outcomes of cardiac dyssynchrony among patients with heart failure attending Benjamin Mkapa Hospital in Dodoma, central Tanzania: A protocol of prospective-longitudinal study

**Patrick Bilikundi[1,2], Baraka Alphonce[1], Azan Nyundo[1,3,4], John Robson Meda**🅾[1,2]*

1 Department of Internal Medicine, School of Medicine & Dentistry, The University Dodoma, Dodoma, Tanzania, 2 Department of Cardiology, The Benjamin Mkapa Hospital, Dodoma, Tanzania, 3 Department of Psychiatry and Mental Health, School of Medicine and Dentistry, The University of Dodoma, Dodoma, Tanzania, 4 Mirembe National Mental Hospital, Dodoma, Tanzania

* jmedaus@yahoo.com, john.meda@udom.ac.tz

**Data Availability Statement:** Deidentified research data will be made publicly available when the study is completed and published.

## Abstract

### Introduction

Cardiac Dyssynchrony is prevalent among patients with heart failure with high cost of care and potentially poor outcomes. Nevertheless, little is known about cardiac dyssynchrony among heart failure patients, especially in developing countries. This study aims at assessing the predictors and outcomes of cardiac dyssynchrony among heart failure patients attending the cardiology department at Benjamin Mkapa Referral Hospital in Dodoma, central Tanzania

### Methods

The study will follow a prospective longitudinal design involving participants aged 18 years and above with heart failure attending the Cardiology Department at Benjamin Mkapa Hospital. Heart failure will be identified based on Framingham's score and patients will be enrolled and followed up for six months. Baseline socio-demographic and clinical characteristics will be taken during enrollment. Outcomes of interest at six months include worsening of heart failure, readmission and death. Continuous data will be summarized as Mean (SD) or Median (IQR), and categorical data will be summarized using proportions and frequencies. Binary logistic regression will be used to determine predictors and outcomes of Cardiac Dyssynchrony among patients with heart failure.

## Introduction

Heart failure is a growing problem globally, resulting in increased mortality and morbidity [1, 2]. Approximately 50% of individuals with heart failure die within 5 yrs of diagnosis [3], while

**Funding:** The author(s) received no specific funding for this work.

**Competing interests:** The authors have declared that no competing interests exist.

yearly mortality is estimated at around 16% [1, 3]. Others end up with recurrent hospitalization and worsening heart failure symptoms [4, 5]. Cardiac dyssynchrony in the background of heart failure makes the outcome worse; up to 60% of individuals with heart failure and Cardiac Dyssynchrony die within four years of diagnosis [6].

Cardiac dyssynchrony is a phenomenon where there is delayed electrical activation in the cardiac myocytes (electrical dyssynchrony) or uncoordinated contraction of the cardiac ventricular walls (mechanical dyssynchrony), which can be either between the segment of cardiac ventricles (intraventricular dyssynchrony) or between the left and right ventricles (interventricular dyssynchrony) [7, 8]

Prevalence of Electrical Dyssynchrony has been found to depend on the aetiology of heart failure [10]. Generally, it is as high as 11% in established heart failure patients, while in dilated cardiomyopathy is estimated to be 72.4% [8, 9]. As for mechanical dyssynchrony, interventricular dyssynchrony has been found to be 79% while intraventricular is 75% [4, 10]. Further studies have reported mechanical dyssynchrony in relation to QRS width, mechanical Dyssynchrony was found to be 20% interventricular and 40% interventricular, in those with QRS $\leq$ 120ms, while it was 58% intraventricular and 76% interventricular among those with QRS $\geq$120ms [11]. Moreover, Nagueh et al in Nigeria found that mechanical dyssynchrony in heart failure patients was as high as 27% for those with narrow QRS complex, while it was as high as 78% for those with wide QRS complex [9]. Additionally, among patients with heart failure; 34% prevalence of mechanical dyssynchrony was found in individuals with QRS $\geq$120 mms, and 11.9% in those with QRS of $\leq$120ms [12]. In another study that was conducted in Ivory Coast, among patients with dilated cardiomyopathy, mechanical Dyssynchrony was estimated to be 70% for intraventricular Dyssynchrony and 47.5% for interventricular Dyssynchrony [13].

Cardiac dyssynchrony in heart failure results in poor outcome, with increased in hospital readmission, worsening of heart failure and death [14]. Recurrent hospitalization and death were some of the outcomes reported, with prevalence of 37% and 19%, respectively [14]

The development of these poor outcomes is multifactorial [5, 13],advanced age, reduced ejection and NYHA class, especially class III and IV are among the key predictors of worse outcomes [15].Furthermore, left ventricle size (dilatation), heart failure etiology, especially dilated cardiomyopathy and ischemic heart disease, poor blood pressure control, diabetes mellitus and electrical dyssynchrony as defined by electrocardiogram are predictors for the development of mechanical dyssynchrony [16]

Studies which were done in Tanzania mainly focused on contemporary aetiology, clinical characteristics and outcomes of HF per se without considering Cardiac Dyssynchrony. For example, in a study which Makubi et al did was assessing the aetiological and common presenting symptoms of patients with heart failure, showed that hypertension was the main aetiology of HF, which accounted for 45% and 48% of the patients had New York Heart Association (NYHA) class III [17].

Despite few studies conducted in Nigeria and Ivory Coast on dyssynchrony in heart failure patients [10, 13, 18], little is still known about cardiac dyssynchrony in our setting, especially its morbidity and mortality, and as a result patients with heart failure and dyssynchrony miss the opportunity for appropriate intervention such as Cardiac resynchronization therapy (CRTDs) [19, 20]. Therefore, this study aims to determine the prevalence, predictors and outcome of cardiac Dyssynchrony among patients with heart failure who will be attending the Cardiology clinic at Benjamin Mkapa Hospital in Dodoma, central Tanzania

The findings of this study will offer convincing justification for developing therapeutic services (CRTDs) in our area, incorporating them into local guidelines, and persuading the health insurance companies to include them in their packages.

## Materials and methods

### Study aims

1. To determine the prevalence of Cardiac Dyssynchrony among adult patients with heart failure attending Cardiology clinic at Benjamin Mkapa Hospital in Dodoma, Tanzania.

2. To determine the predictors of cardiac Dyssynchrony among patients with heart failure attending Benjamin Mkapa Hospital in Dodoma, Tanzania.

3. To determine the six-month outcomes of patients with heart failure and cardiac Dyssynchrony attending Benjamin Mkapa Hospital in Dodoma, Tanzania.

### Study design

A prospective longitudinal observational study will be conducted among adult patients with heart failure attending cardiac clinic at Benjamin Mkapa hospital.

### Study area

The study will be conducted in the cardiology department at the Benjamin Mkapa Hospital (BMH), located in the city of Dodoma, 16 kilometres from the city centre. BMH serves as a consultant hospital for the Central zone and other neighbouring areas of Dodoma. The hospital serves around 11.3million population according to 2022 TDHS, the majority being from Dodoma region, about 5.68million; while the rest are from nearby regions, including Singida, Manyara, Tabora, Morogoro, and Iringa [21]. Moreover, it is used as a teaching hospital for the University of Dodoma (UDOM). Advanced radiological tests, including CT scans, MRIs, cardiological diagnostic tests, and interventional procedures like Coronary Angiography, Percutaneous Coronary Intervention, and cardiac pacing are available at Benjamin Mkapa Hospital

BMH has a bed capacity of 400, and approximately 240 patients with heart conditions usually attend the cardiology department per month, of whom 15–20 patients are of heart failure and six patients with Cardiac Dyssynchrony are registered per week (Unpublished data).

### Sample size estimation

In order to estimate the sample size, the proportionate formula for prospective cohort studies will be used [22]

$$n = \frac{\left(Z_{\alpha/2}\sqrt{\left(\frac{r+1}{r}\right)p*(1-p*)} + Z_{\beta}\sqrt{\frac{p_1(1-p_1)}{r} + p_2(1-p_2)}\right)^2}{(p_1 - p_2)^2}$$

$$p* = \frac{p_2 + rp_1}{r+1}$$

Where by
r = ratio between the two groups
p1 = Cardiac dyssynchrony prevalence obtained from literature
p2 = expected prevalence from the study
p1 −p2 = effect size
$Z_{\beta}$ = standard normal variate for statistical power

$Z_{\propto/2}$ = standard normal variate for significance level

The prevalence of cardiac dyssynchrony in previous studies was 12.9%

The expected prevalence in this study is expected to be 16%

Therefore;

r = 1

p1 = 12.9%

p2 = 16%

p1 −p2 = -3.1%

$Z_\beta$ = 0.84 for statistical power of 80%

$Z_{\propto/2}$ = 1.96 for significance level of 95%

N = 134

Considering the 20% attrition rate in our setting

Therefore, the minimum sample size estimated is 150 patients.

## Inclusion criteria

1. All patients aged ≥18 years with signs and symptoms of heart failure based on the Framingham score

2. All patients who will be able to provide informed consent, and for those who will be un able a surrogate relative will consent on their behalf [23].

## Exclusion criteria

1. All patients with heart failure aged ≥18 years with LVAD.

2. All patients with heart failure aged ≥18 years with a functional pacemaker

3. All patients with heart failure aged ≥18yrs with ventricular rhythms.

4. All patients with advanced malignancies or end stage medical conditions

## Sampling methods/techniques

Consecutive Sampling method will be used whereby the sample will be attained by selecting every participant who fits the inclusion criteria until the desired sample size is reached.

## Assessment of heart failure

A detailed history and general physical examination will be conducted on all participants. The diagnosis of heart failure will be reached based on Framing Harm criteria; two major criteria, or one major and two minor criteria [24]. Major criteria will include: Paroxysmal nocturnal dyspnoea, Neck vein distention, Rales (crackles in the lungs) Cardiomegaly (enlarged heart) on chest X-ray Acute pulmonary oedema S3 gallop (a third heart sound). Minor criteria will include: Oedema, Nocturnal cough, Hepatomegaly (enlarged liver), Pleural effusion (fluid accumulation around the lungs) and Dyspnoea on exertion

## Assessment of cardiac dyssynchrony

**Electrical dyssynchrony.** A standard 12-lead surface electrocardiogram (ECG) machine (GE Healthcare MAC$^{TM}$ 2000 USA; 2020) set at a paper speed of 25 mm/s and a scale of 10 mm/mV will be used to assess electrical Dyssynchrony. The procedures for performing ECG

will follow the standard operating procedure set by the cardiology department of the Benjamin Mkapa hospital. This protocol was adopted and customized from the established guidelines of the American Heart Association [25] Participant will be asked to expose the chest adequately and positioned supine on the cardiac bed. There will be no need to shave the skin as long as wet gel electrodes are used. Precordial leads will be six; V1-V6, V1 and V2 will be placed along the fourth intercostal space, 2cm right and left side of the sternum respectively. V4 will be placed in the fifth intercostal space, while V3 will be placed in between V2 and V4. V5 and V6 will be placed at fifth intercostal space, along the medial axillary line and axillary line respectively. The limb electrodes will be applied as follows: Red on the right inner wrist, yellow on the left inner wrist, black just above the ankle on the right inner leg, and green just above the ankle on the left inner leg. After entering the patient's ID number into the device, a device will be started to analyze the electrical activity and a printout will be obtained. Electrical dyssynchrony will be ascribed to those with a widened RSS complex of more than 120ms [5]. Interpretation of the findings will be done by the principal investigator and confirmed by two experienced cardiologists. Any disagreement will be settled in agreement with a third independent experienced cardiologist.

**Assessment of mechanical dyssynchrony.** Mechanical Dyssynchrony will be assessed using Transthoracic echocardiography (model Vivid $^{TM}$ T9 made by GE Healthcare, USA 2018). The patient will be asked to expose the chest and abdomen above the symphysis pubis and positioned in the left lateral position. ECG tracing rhythm leads will be placed on the chest to pick the QRS complexes timing the onset of contraction corresponding to the aortic and pulmonic valvular closure. A two-dimensional (2D), M-mode, Doppler echocardiogram and speckle tracking echo will be performed; parasternal, short axis, apical four-chamber views and five chamber view. The dimensions of the left atrial, LV, and other cardiac chambers will be determined using a 2D guided M-mode approach. The left ventricle pre-ejection interval (LPEI), right ventricular pre-ejection interval (RPEI), and septal to posterior wall motion delay will be assessed [26].

The time between the start of the QRS complex and the start of the aortic flow velocity detected in the apical five-chamber image will be used to compute the aortic pre-ejection time, termed as LPEI. From the start of the QRS complex until the start of the pulmonary flow velocity curve visible in the left parasternal view, the pulmonary pre-ejection time is calculated, termed as RPEI. LPEI will be defined as the distance measured from the start of aortic Doppler flow velocity curve to the start of QRS, while RPEI will be defined as the distance measured from the start of the pulmonary doppler flow velocity curves to the start of the QRS. A 40ms difference between LPEI and RPEI will be considered as inter ventricular mechanical delay (IVMD). The time interval between the maximal outward displacement of the inter ventricular septum and the LV posterior wall will be measured as a septal to posterior wall motion delay (SPWMD) [27] and a value of >130 ms 2 standard deviation above the mean of normal controls will be considered the presence of intraventricular dyssynchrony [9]. Intraventricular dyssynchrony will be defined as LPEI 140 ms in patients with difficult-to-identify systolic peaks of the LV posterior wall or an akinetic interventricular septum [28].

## Outcome measurement

**Primary outcome.** The primary outcome of interest will be the presence of cardiac dyssynchrony. Cardiac dyssynchrony will be either mechanical, electrical or both

**Mechanical dyssynchrony - An** echo will be performed to evaluate the delay in contractility in ventricular chambers between the chambers (interventricular) or between segments of the left ventricle (intraventricular)

Electrical dyssynchrony- a standard 12 ECG leads will be used to evaluate the width of QRS complex, where anyone with value of $\geq$120ms will be termed as having electrical dyssynchrony.

## Secondary outcomes

**Worsening of heart failure.** During the follow-up period, a patient with cardiac dyssynchrony and chronic heart failure on optimal medical tolerable therapy with no symptoms (stable) will be termed as having worsening of heart failure if signs and symptoms of decompensated heart failure develop and necessitates either escalation of oral diuretics dose or iv administration to relieve symptoms in the outpatient setting [29].

**Readmission for heart failure symptoms.** During follow-up, any worsening of heart failure symptoms in a patient with chronic heart failure and cardiac dyssynchrony that warrant in hospital management to relieve symptoms using diuretics and venodilators will be assessed as readmission for heart failure symptoms [29].

**Mortality.** *Cardiovascular death.* Will be defined as any death that occurs following hospitalization due to heart failure or any sudden death with no any explainable cause in heart failure patients with cardiac dyssynchrony.

*All-cause mortality.* Any death occurring in a patient with heart failure and cardiac dyssynchrony hospitalized for any other disease condition apart from heart failure or death without an identifiable cause.

## Participants' characteristics

Participants will be adults aged 18 years or older who will be attending the cardiology department of the Benjamin Mkapa hospital with established diagnosis of heart failure using the framing harm criteria (two major criteria or one major and two minor) as described in the assessment of heart failure section.

## Data collection process

**Evaluation of the participants.** A total of 150 participants who will meet the inclusion criteria will be enrolled from July 2023. Participants who consent will be interviewed using a structured questionnaire that collects socio-demographic data such as age, gender, and area of residence. A minimum of two contacts; from the participant and next of kin, will be inquired. Moreover, alcohol consumption history will be taken with its duration (current drinker will be defined as alcohol consumption within the past 12 months, or currently taking alcohol while formerly used alcohol will be defined as someone who lastly took alcohol past 12months and currently doesn't take alcohol) [1, 30]. Cigarette smoking history will be taken and reported in cigarettes packs per year, the history of current smoking, will be defined as smoking within the past year or currently active smoking, while past smoker will be defined as someone who last smoke 12month ago and currently doesn't smoke [1]. Information regarding potential predictors of cardiac dyssynchrony will also be collected, these include; heart failure symptoms functional class, presenting ejection fraction and conduction delay will be taken [31]. Also other clinical characteristics including history of Hypertension (defined as a history of Hypertension or the use of antihypertensive medications) and diabetes (defined as a history of diabetes or the use of diabetic medications) will be documented [32].

## Determination of the predictors of cardiac dyssynchrony

**Ejection fraction.** Using an echo machine model Vivid $^{TM}$ T9 made by GE Healthcare, USA 2018), a transthoracic echo will be performed in parasternal long axis view, using 2D

mode, left ventricular end diastolic volume (LVEDd) at the end of diastole will be measured, as well as the left ventricular end systolic volume (LVESD) at the end of systole. Ejection fraction (EF) will then be computed as the stroke volume (given as LVEDD-LVSED) divided to LVEDD then multiplied by 100 to give the results in percentages. EF will be termed as reduced when its 50% and above, while a value below 50% will be termed as reduced.

**Age.** Information about the age in years will be obtained from the medical records, and ascertainment of the information will be done by asking the participants.

**Sex.** The information regarding sex of participants will be obtained from the medical records

**Etiology of heart failure.** Using the echo machine model Vivid [TM] T9 made by GE Healthcare, USA 2018, a transthoracic echo will be performed to participants for determination of etiologies of heart failure.

Hypertensive heart disease (HHD) will be assigned to the participants with history of hypertension who upon doing an echo on long axis parasternal view, under m-mode the dimensions of interventricular septum thickness and left ventricular thickness are more than 11mm.

Dilated cardiomyopathy will be assigned to a participant with increase diameter of the ventricular size of more than 5.6cm during diastole recorded at apical four chamber view, and a thickness of interventricular septum and ventricular wall thickness of less than 0.6cm taken at m mode

Ischemic heart disease: under 2D short axis parasternal view the segments of the left ventricle will be visualized. A participant who will be having a regional wall motion abnormality will suffice to be diagnosed of having ischemic heart disease [33]

**New york heart association (NYHA).** The NYHA is a clinical classification for severity of heart failure symptoms, assessed through asking tolerance to physical activities. The principle investigator will inquire the information on participants about tolerance to physical activities and classify participants according to American Heart Association (AHA) functions classification as follows Class I: No limitation of physical activity, Class II: Slight limitation of physical activity, Class III: Marked limitation of physical activity, Class IV: Inability to carry out any physical activity without discomfort [34].

**Diabetes mellitus (DM).** Those who are on medication for diabetes mellitus from the medical history, and those who are not with signs and symptoms of Hyperglycermia with fasting glucose level of 7.1mmol/litre will be termed as having diabetes mellitus [35]

**Body mass index (BMI).** The BMI will be computed from heigh from the height and weight of the patient, values given in kilogram per metre square, and classification will follow World Health Organization as follows; underweight as BMI of <18.5kg/m2, normal BMI 18.5–24.9kg/m2, overweight as 25–29.9kg/m2, and obese as BMI as ≥30kg/m2 [36]. The procedures for measuring and calculating the BMI will follow that described by Jessica et al as follows [37]

Weight. The patient will be asked to wear light clothing and no shoes. The weight will be measured using a Seca weighing scale from German that is calibrated in kilograms. The patient will stand on the scale with their feet slightly apart and their weight evenly distributed.

Height. To measure height accurately, the patient will be asked to stand barefoot with their back straight against a flat surface of a stadiometer. The head, shoulders, buttocks, and heels are kept in contact with the surface, while the patient is looking straight ahead. The height is then measured from the top of the head to the floor machines by using vertical measuring rod with a sliding horizontal headpiece that is used to determine the patient's height. The Seca 213stadiometer from German will be used.

## Clinical variable measurement

**Blood pressure measurement.** A blood pressure (BP) reading will be measured using an automated digital machine of the AD Medical Inc. brand, Model UA-611 with a Slimfit™ cuff of United Kingdom (2009) [38] keeping with the 2018 AHA/ACC Hypertension guideline for standard measurement of BP [39]. Hypertension will be defined as BP $\geq$140/90 mmHg after three different reading measured at least 5 minutes apart or a patient with a known history of hypertension or a patient on antihypertensive medications [40].

**Radial pulse measurement.** Two operators will assess the radial pulse using a timer for one minute. The arm of the participant will be slightly pronated and the wrist slightly flexed [41]. Radial pulses will be palpated against the radial head using the ring, middle and index fingers, in supinated [41]; To eliminate bias, the two operators will measure separately, and if they get separate values, an average of the two will be taken and us [42]

## Study variables

**Aim 1 Study Variables:** This variable addresses the prevalence of cardiac Dyssynchrony among patients with heart failure. The outcomes include electrical dyssynchrony that will be measured using an ECG assessing the width of the QRS complex. and mechanical dyssynchrony that will be assessed using echocardiography, both measured in milliseconds. Summary of description of aim 1, see Table 1 below.

**Aim 2 Study Variables:** These will assess the predictors of Cardiac dyssynchrony among heart failure patients. Age measured in years, ejection fraction measured in percentage, sex given as male or female, BP status given as high, low or normal, BMI given as obese or non-obese, heart failure aetiology and symptoms of heart failure assessed using New York Heart Association Functional class. Summary of description of aim 2, see Table 2 below.

**Aim 3 study variables:** these variables will assess the outcomes; readmissions, worsening of heart failure and death. Summary of description of aim 3, see Table 3 below.

## Data management

Data will be collected using a structured questionnaire that was adopted and customized from a tool that was used in a multi-national heart failure registry; The global congestive heart failure (GCHF) [1]. Each participant will be assigned a unique anonymous identifier number. Data will be stored on password-protected and encrypted computers to maximize confidentiality and security. Anyone who will need to access the data for secondary data analysis an approval will be given by the authorized research board of the institution where the data will be stored in consultation with the primary researcher of this study.

**Table 1. Summary and description of objective number 1 of the study.**

| variable | Method of measurement | Operational Definition of variable | Level of measurement |
|---|---|---|---|
| Cardiac dyssynchrony | Mechanical dyssynchrony will be assessed and measured using echo, while electrical dyssynchrony will be measured using 12 lead ECG | Cardiac dyssynchrony will be either mechanical, electrical or both **Mechanical dyssynchrony—will** be termed as interventricular if the difference between LPEI and RPEI is 40ms. Intraventricular will be defined by the interval between the maximal outward displacement of the septum and the LV posterior wall delay being more than 130ms. **Electrical dyssynchrony**- a standard 12 ECG leads will be used to evaluate the width of QRS complex, where anyone with a value of $\geq$120ms will be termed as having electrical dyssynchrony | Dichotomous |

**Table 2. Summary of description of predictors of outcome.**

| Variable | Method of measurement | Operational definition | Level of measurements |
|---|---|---|---|
| Age | Counting the number of years from birth date (obtained from medical records) up to the date of enrollment. | Age in years | Continuous |
| Ejection fraction | Will be measured using ECHO. | Percentage of blood pumped out of the left ventricle after every beat | Continuous |
| Sex | Will be self-reported or inquired from medical records. | Male or female | Dichotomous |
| BMI | Will be calculated after measuring height using a stadiometer and weight using a calibrated weighing scale | Underweight $<18.5\text{kg/m}^2$<br>Normal $18.5–24.9\text{kg/m}^2$<br>Overweight $25–29.9\text{kg/m}^2$<br>Obese $\geq30\text{kg/m}^2$ | ordinal |
| BP status | Medical records, self-reporting or assessment | High if $\geq140/90$mmhg<br>Hypotensive if $<90/60$<br>Normal if any vale in between these readings | ordinal |
| Heart failure aetiology | ECHO | HHD: IF history of chronic hypertension or evidence of left ventricle hypertrophy on echo<br>IHD: if history of CAD, evidence of regional wall motion abnormalities on echo.<br>DCM: if evidence of left vertical dilatation on echo<br>Co pulmonale: if there's coexistence of pulmonary diseases and right sided heart failure | nominal |
| Heart failure functional class | Symptoms: NYHA | **Class I (mild):** no symptoms with ordinary activities<br>**Class II (moderate):** symptoms on moderate physical activity<br>**Class III (moderate to severe):** symptoms on mild activities<br>**Class IV (severe):** symptoms even at rest | ordinal |

## Data analysis

Data will be entered on a Microsoft Excel sheet for statistical analysis and then converted to IBM SPSS PC version 26 [43]. Continuous variables will be summarized as mean and standard deviation (SD), or median and interquartile ranges, for non-parametric variables; frequencies and percentages will be used for categorical variables. Descriptive statistics will be used to summarize demographics and clinical characteristics using frequencies and proportions. Binary logistic regression will be used to determine the predictors of outcome, and crude odds ratio will be used as a measure of association. The predictors for multivariable logistic regression will be those that are statistically significant in univariate logistic regression, those that literatures support that they have association with the outcome, or whose p value is $\leq0.2$. [44] These will include ejection fraction, male sex, NYHA, BMI, DM and BP status; and adjusted odds ratio will be used as a measure of association. At the same time, 95% confidence interval (CI) will be calculated concurrently and a two-sided $p\leq0.05$ will be used to establish statistical significance.

**Table 3. Summary of description of outcomes.**

| Variable | Method of measurement | Level of measurement |
|---|---|---|
| Worsening of heart failure | Prior stable patients who develop signs and symptoms of heart failure and either need:<br>Escalation of diuretic dosage<br>IV diuretics administration to relieve symptoms | Dichotomous |
| Readmission | Any event of inpatient management of heart failure symptoms due to either decompensation from known trigger or worsening of heart failure | Dichotomous |
| Death | Death following admission from heart failure symptoms, death occurring while seeking management of heart failure symptoms or any sudden death without any explainable cause | dichotomous |

## Ethical issues

The Vice Chancellor's office at the University of Dodoma provided permission to conduct the study after obtaining ethical clearance from the Directorate of Research and Publications with reference number MA.84/261/02/'A'/59/14. The Institutional Research Review Ethics Committee (IRREC) convened a meeting to address the ethical issues pertaining this study Following that, the administrative department of Benjamin Mkapa approved the data collection with reference number AB.150/293/01/298. A written informed consent will be obtained from participants for inclusion in the study. Participants will be informed that their participation is completely optional and that they might withdraw at any time. After data collection, participants' identities will be replaced with identification numbers in order to maintain privacy and confidentiality. Those with cardiac desynchrony will be channelled to appropriate centres capable of resynchronization therapy and any other appropriate management that might benefit them.

**Data availability and dissemination of results.** Data will be made available at the University of Dodoma and Benjamin Mkapa hospital repository. Data will be accessible after consulting the primary investigator.

The study's findings are expected to be published in peer-reviewed journal the PLOS ONE with the aim of communicating these findings to the entire world to improve patient outcomes. Furthermore, these study findings will further be presented to the Ministry of Health to equip them with a basis for incorporating CRTDs as a must therapeutic service in our local guidelines to manage patients with heart failure and cardiac dyssynchrony.

## Study timeline

The study is expected to take twelve months; three months is expected to suffice participant enrollment, six months of follow-up and the remaining three months for data analysis and reporting of results.

## Discussion

The presence of CD in HF patients has been linked to increased mortality rates and recurrent hospital admissions [45]. Studies have reported a 5-year overall survival rate of only 50% in HF patients [3] and a mortality rate of 68% in patients with HF and cardiac dyssynchrony after a 4-year follow-up [6]. Despite the introduction of novel pharmacologic medications like beta-blockers and angiotensin-converting-enzyme inhibitors (ACEIs) in HF management, the quality of life for these patients remains poor with the majority of them occurring as early as within 6-month [45, 46]. CD exacerbates the risk of death and worsens the survival outcomes in HF patients [9]. The study highlights the significant impact of cardiac dyssynchrony (CD) on patients with heart failure (HF) and its association with poor prognosis and reduced overall survival [45]. It emphasizes the need for timely diagnosis and appropriate treatment of CD to improve patient outcomes and quality of life. Interestingly, studies have shown that even though these people have severe heart failure, if they receive cardiac resynchronization therapy (CRT), they do better, have a lower risk of recurrent hospitalization, and lower mortality rate [9, 27].

However, it can be challenging to stay in touch with every cohort member in a prospective cohort research, especially if the cohort is sizable and the time of follow-up is lengthy. When numerous patients are lost to follow-up, the results may be skewed, particularly when the risk factor or outcome of interest is connected to the reason for the loss to follow-up. Furthermore, the ability of a cohort study to deduce causality from the observed link between cardiac dyssynchrony and bad outcomes is limited.

The study site, Benjamin Mkapa Hospital, and the University of Dodoma library will get the final results, and a manuscript will be prepared for submission to a number of peer-reviewed journals prior to publication.

## Acknowledgments

I would like to express my gratitude to my supervisors, Dr. John R. Meda, Dr. Azan Nyundo, and Dr. Baraka Alphonce for their tireless and endless guidance and support throughout the development of this study protocol. They had a significant role in forming the study topics and techniques with their work, knowledge, and insight. They have been steadfast in their dedication to assuring the accuracy and reliability of this study, and I appreciate how eager they have been to offer suggestions and support along the road. They provided the direction that made this protocol practicable.

## Author Contributions

**Conceptualization:** Patrick Bilikundi, John Robson Meda.

**Data curation:** Patrick Bilikundi.

**Formal analysis:** Patrick Bilikundi.

**Investigation:** Patrick Bilikundi, John Robson Meda.

**Methodology:** Patrick Bilikundi, John Robson Meda.

**Supervision:** Azan Nyundo, John Robson Meda.

**Writing – original draft:** Patrick Bilikundi, Azan Nyundo, John Robson Meda.

**Writing – review & editing:** Patrick Bilikundi, Baraka Alphonce, Azan Nyundo, John Robson Meda.

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
