## [Decision Letter · Decision Letter 0]

19 Sep 2023

PONE-D-23-16923Predictors and outcomes of Cardiac Dyssynchrony among patients with heart failure attending Benjamin Mkapa Hospital in Dodoma, central Tanzania: A protocol of prospective-longitudinal study.PLOS ONE

Dear Dr. Meda,

Thank you for submitting your manuscript to PLOS ONE. After careful consideration, we feel that it has merit but does not fully meet PLOS ONE’s publication criteria as it currently stands. Therefore, we invite you to submit a revised version of the manuscript that addresses the points raised during the review process.

Please address the concerns raised by reviewers in comments appended below before we can consider your manuscript further.

We look forward to receiving your revised manuscript.

Kind regards,

Yashendra Sethi

Academic Editor

PLOS ONE

Journal Requirements:

Additional Editor Comments (if provided):

Thank you for submitting the work for consideration with PLOS One. The manuscript has now been evaluated by the subject expert reviewers.The topic is of potential interest but the reviewers have underlined some major concerns which need to addressed. The reviewer comments appended below. Kindly revise as per the comments of reviewers. Since, we are considering protocols too now, you may skip the comments of respected reviewer R2 regarding the journal criterion for protocol as a study design, however other concerns need to ne addressed before we can consider your work further.

Reviewers' comments:

Reviewer's Responses to Questions

**Comments to the Author**

1. Does the manuscript provide a valid rationale for the proposed study, with clearly identified and justified research questions?

Reviewer #1: Yes

Reviewer #2: Yes

2. Is the protocol technically sound and planned in a manner that will lead to a meaningful outcome and allow testing the stated hypotheses?

Reviewer #1: Yes

Reviewer #2: Yes

3. Is the methodology feasible and described in sufficient detail to allow the work to be replicable?

Reviewer #1: Yes

Reviewer #2: Yes

4. Have the authors described where all data underlying the findings will be made available when the study is complete?

Reviewer #1: Yes

Reviewer #2: No

5. Is the manuscript presented in an intelligible fashion and written in standard English?

Reviewer #1: Yes

Reviewer #2: Yes

6. Review Comments to the Author

You may also provide optional suggestions and comments to authors that they might find helpful in planning their study.

Reviewer #1: Predictors and outcomes of Cardiac Dyssynchrony among patients with heart failure

attending Benjamin Mkapa Hospital in Dodoma, central Tanzania: A protocol of

prospective-longitudinal study

General:

Heart failure is a growing problem worldwide and in Tanzania its prevalence is estimated to be about 12 % in the hospital setting and is believed to be on the increase. Presence of cardiac dyssynchrony in heart failure has been associated with poor outcome, with increased in hospital readmission, worsening of heart failure and death Therefore, this protocol is important as will help provide data that are relevant and of great public health interest.

The rationale is straight forward. The objectives of the study are clear. For the most part the methods are appropriate and well described. The author further summarized the limitation of the study. Despite these strength, there are some aspects of the manuscript the needs to be addressed.

Abstract

1. This is well written and contains all the elements of a good abstract, but minor corrections to make: the word “yrs” to be re-written as “years” and a full stop to be added after the last sentence.

Introduction:

2. This is also well written, however rearrangement of flow is required for example the sentence that says “This will offer convincing justification for developing therapeutic services (CRTDs) in our area, incorporating them into local guidelines, and persuading the health insurance companies to include them in their packages” that seems to be a justification of the study thus need to come after the last paragrapgh and re-written as “The finding of this study will offer…”. Also a reference is required for this statement: “Despite few studies conducted in Nigeria and Ivory Coast on dyssynchrony in heart failure……”

3. Unfortunately, the manuscript was not numbered line-by-line in order to provide simple peer-review, I recommend the author to do this for re-submission.

Methods:

By and large this section is okay, except for minor comments under the following parts:

Sample size estimation

5. The author should avoid repetition of words in same sentences i.e the word “used”

Inclusion criteria

6. The author should use appropriate tense, for example a statement; “All patients who will be able to provide informed consent, and for those who will be unable, a surrogate relative will consent on their behalf.

7. Is it ethically allowed for surrogate relative to consent for a patient who unable to consent by themselves.

Assessment of Cardiac dyssynchrony

8. Additional information is required to get flow for the reader to follow: for example the statement; “After entering the patient's ID number into the device, the device will be started and a printout obtained

Assessment of Mechanical dyssynchrony

9. LINE 5: Valvular “closer “ to be changed to “closure”

10. LINE 14: Use of appropriate tense; “The time between the start of the QRS complex and the start of the aortic flow velocity detected in the apical five-chamber image will be used to compute the aortic preejection time, termed as LPE

Data collection process

11. The author need to show at which stage data such as BMI (obesity status are obtained (i.e from file by using questionnaire/measurement) and also how BMI will be categorized.

12. I recommend the author to amend the flow in the methodology for the Study variables ( study aims and summary tables for all aims to come after Data collection process and before Data management, to allow good flow and easiness of reader to follow.

Evaluation of participants

13. LINE 11: the author need to show from where and how the information regarding potential predictors of cardiac dyssynchrony will be obtained.

Clinical Variable measurement

Blood pressure measurement

14. The author need to specify the detail of the automated BP machine to be used i.e model, company, country of origin, and year)

Data analysis

15. The author should add a reference for the Statistical software (IBM SPSS PC version 26) to be used.

Discussion

16. LINE 11: Reference is required for this statement “Interestingly, studies have shown that even though these people have severe heart failure, if they receive cardiac resynchronization therapy (CRT), they do better, have a lower risk of recurrent hospitalization, and lower mortality rate.”

Reviewer #2: Thank you for the opportunity to review the manuscript titled, "Predictors and outcomes of Cardiac Dyssynchrony among patients with heart failure

attending Benjamin Mkapa Hospital in Dodoma, central Tanzania: A protocol of

prospective-longitudinal study". I appreciate the effort put by Dr. Meda et al. in addressing an important issue related to cardiac dyssynchrony among heart failure patients.

However, after careful consideration, I must point out that the manuscript does not include the reporting of results and appears to be a only a protocol of an ongoing study. As per the guidelines set by PLOS One, manuscripts submitted for publication should include complete and meaningful results. The current state of the manuscript does not align with these guidelines.

While the research question and significance of the study are evident, I recommend that they consider revising the manuscript once they have gathered and analyzed the results of their ongoing study or reported the result of a pilot study conducted for the protocol. Including concrete findings and outcomes will make the manuscript more suitable for publication in the journal.

Thank you for considering my feedback, and I look forward to potentially reviewing the revised manuscript in the future.

7. PLOS authors have the option to publish the peer review history of their article (what does this mean?). If published, this will include your full peer review and any attached files.

Reviewer #1: No

Reviewer #2: No

---

## [Author Response · Author response to Decision Letter 0]

25 Oct 2023

(please see the revised manuscript with track changes for reference )

1. To change “yrs” to “years” 

response: The word has been rewritten into a full form “years”. see Page 2; Line 32

2. Rearranging the paragraph of the rationale “This will offer convincing justification for developing therapeutic services (CRTDs) in our area, incorporating them into local guidelines, and persuading the health insurance companies to include them in their packages” to be the concluding one 

response: The statements/paragraphs have been rearranged and written properly in chronology. see Page 4; Line 88-93

To put references for the statement “Despite few studies conducted in Nigeria and Ivory Coast on dyssynchrony in heart failure……”. 

response: The reference has been added to the statement see Page 4; Line 85

3.The reviewer wished if the manuscript could be numbered from the first draft 

 response: The numbering and track changes has been made available in the following versions

5.To avoid repetition of the words, especially “used” in sample size estimation 

response: The word has been omitted and a statement written correctly see Page 6 Line 123

6. Using the appropriate tense in writing the protocol, a future tense to be observed. 

response: All patients who will be able to provide informed consent, and for those who will be unable, a surrogate relative will consent on their behalf. see Page 7 Line 156

7 Reviewer wanted clarification if it is ethically allowed for surrogate relative to consent on behalf for those who will be un able? 

response: In this protocol we abided to “The Mental Capacity Act 2005” provides a legal framework for acting and making decisions on behalf of adults who lack the capacity to make particular decisions for themselves. see Page 7 Line157 Reference number 24

8 Reviewer wanted additional information to be added on assessment of electrical dyssynchrony to get a good chronological flow of information 

response: the information has been added on the device operation on assessing electrical dyssynchrony. see Page Line 190-191

9. The word “closer” to be changed to “closure” in assessment of mechanical dyssynchrony. 

response: The word has been changed see Page 8 Line 201

10. Rewriting the sentence in appropriate tense 

response: The sentence was changed to “The time between the start of the QRS complex and the start of the aortic flow velocity detected in the apical five-chamber image will be used to compute the aortic preejection time, termed as LPEI see Page 8 Line 208

11. The reviewer suggested showing at which stage data such as BMI (obesity status are obtained (i.e from file by using questionnaire/measurement) and also how BMI will be categorized. 

response: The information has been added on the part of methodology on assessing the predictors of cardiac dyssynchrony. see Page 13, Line 328-342

12. Reviewer suggested the flow of writing to be amended for the Study variables (study aims and summary tables for all aims to come after Data collection process and before Data management, to allow good flow and easiness of reader to follow.

response: The flow has been amended . see Page 14-16 Line 358-380

13. The reviewer wanted the information regarding the predictors of cardiac dyssynchrony where and how will be obtained 

response: How and where the information regarding predictors of cardiac dyssynchrony will be obtained has been added to the manuscript. Page 12-13, line 391-342

14. To specify the detail of the automate BP machine to be used 

response: The specifications have been added on the manuscript see Page 13, line 347

15. To add reference for the statistical software that will be used i.e IBM SPSS PC version 26 

response: The reference has been added. see Page 17, line 392

16. References regarding the information “Interestingly, studies have shown that even though these people have severe heart failure, if they receive cardiac resynchronization therapy (CRT), they do better, have a lower risk of recurrent hospitalization, and lower mortality rate” to be added 

response: The references have been added. see Page 19, line 449

---

## [Editor Report · Decision Letter 1]

3 Nov 2023

Predictors and outcomes of Cardiac Dyssynchrony among patients with heart failure attending Benjamin Mkapa Hospital in Dodoma, central Tanzania: A protocol of prospective-longitudinal study.

PONE-D-23-16923R1

Dear Dr. Meda,

We’re pleased to inform you that your manuscript has been judged scientifically suitable for publication and will be formally accepted for publication once it meets all outstanding technical requirements.

Kind regards,

Yashendra Sethi

Academic Editor

PLOS ONE

Additional Editor Comments (optional):

Thank you for addressing all the comments. I feel happy to recommend the revised version for publication.
---

## [Editor Report · Acceptance letter]

8 Nov 2023

PONE-D-23-16923R1 

Predictors and outcomes of Cardiac Dyssynchrony among patients with heart failure attending Benjamin Mkapa Hospital in Dodoma, central Tanzania: A protocol of prospective-longitudinal study. 

Dear Dr. Meda:

I'm pleased to inform you that your manuscript has been deemed suitable for publication in PLOS ONE. Congratulations! Your manuscript is now with our production department. 

Kind regards, 

on behalf of

Dr. Yashendra Sethi 

Academic Editor

PLOS ONE